# Lost in Translation: Tangible and Non-Tangible in Conservation

Nigel Walter

Department of Archaeology, University of York, York YO1 7EP, UK; nhw502@york.ac.uk

**Abstract:** This paper addresses the special issue theme of the response of conservation practice to shifts in heritage theory towards the intangible, through exploring some specific aspects of practice and statutory process in the UK. The paper starts with an overview of conservation in the UK, and the extent to which it does or does not interface with developments in heritage theory. It explores the conventional understanding of significance—here termed 'subtractive'—which reflects the antiquarian concerns from which conservation developed. It then considers the Ecclesiastical Exemption, a parallel consent mechanism within UK law for Christian places of worship that remain in use, which specifically recognises their need to change over time to ensure their survival. Evidence for a growing appreciation of non-tangible value and community participation in heritage is provided in recent research by The National Churches Trust into the economic and social value of church buildings to local communities across the UK. The paper concludes that a positive response to changes in heritage theory requires conservation to undertake its own theoretical work; this will involve a recognition of living buildings as central rather than peripheral both to conservation and to heritage more broadly, and a move towards a 'generative' understanding of significance.

**Keywords:** conservation; intangible heritage; Ecclesiastical Exemption; Faculty Jurisdiction; significance; National Churches Trust; heritage theory

## 1. Introduction

This paper addresses the special issue theme of the response of conservation practice to the shift of heritage theory from its former focus on tangible product towards an understanding of intangible process. It does this through exploring some specific aspects of practical experience of the conservation of living buildings in the UK, particularly historic church buildings under the Ecclesiastical Exemption; the parallel issue of the adaptive reuse of historic church buildings no longer in use raises a complementary but distinct set of issues and is not addressed.

For the purposes of the paper, tangible heritage can be defined as material forms of heritage, including historic buildings, monuments, artefacts, artworks, archaeological remains, etc. The labelling of these as 'tangible' flowed from the recognition in the later twentieth century of other forms of heritage that are 'intangible' [1–3], a shift that culminated in the 2003 Convention for the Safeguarding of the Intangible Cultural Heritage, which defines intangible cultural heritage as 'the practices, representations, expressions, knowledge, skills [. . .] that communities, groups and, in some cases, individuals recognize as part of their cultural heritage' ([3], art. 2.1). This is understood to be 'transmitted from generation to generation', and is 'constantly recreated' by communities and groups, moving the focus of heritage away from a physical product towards an intergenerational process.

The nature of the interrelation of these two forms of heritage has been a cause of concern from the outset; in 2004, the *Yamato Declaration on Integrated Approaches for Safeguarding Tangible and Intangible Cultural Heritage* noted their interdependence and called for the elaboration of 'integrated approaches', without exploring the nature of their interrelation [4]. A few weeks later, the World Heritage Committee noted the relevance of the *Declaration* but stressed that the two conventions 'address different forms of heritage

and therefore [. . .] have different scopes' [5]. The resulting understanding—at least in a World Heritage context—is of two domains of heritage which overlap, but which remain fundamentally dissimilar.

This paper explores this relationship between the tangible and, using the earlier and now more neutral term, the 'non-tangible'; it aims to question the conventional distinction between the two, arguing instead for continuity between them.

## 2. Conservation Practice in England and Beyond

In a UK context, Historic England is the national government's statutory adviser on the historic environment in England; central amongst its varied range of guidance for heritage professionals is its 2008 document *Conservation Principles, Policies and Guidance* [6]. Notwithstanding a controversial and ill-fated attempt at revision in 2018, this remains the organisation's core guidance document [1]. In methodology, it adapts the *Burra Charter's* fourfold structure of values, and adopts the language of people and place [7]. It defines conservation as:

> the process of managing change to a significant place in its setting in ways that will best sustain its heritage values, while recognising opportunities to reveal or reinforce those values for present and future generations ([6], p. 7).

This placing of the 'management of change' at the centre of conservation has far-reaching consequences, and remains a significant issue for some heritage professionals, because it challenges the previously dominant preservationist understanding, that responsible conservation necessarily involves constraining change to the minimum. By contrast, Historic England declares an openness to change, for example noting that:

> The concept of conservation area designation, with its requirement 'to preserve or enhance', also recognises the potential for beneficial change to significant places, to reveal and reinforce value ([6], p. 15).

Clearly, conservation areas are subject to a different form of protection from that of individual buildings; nevertheless, there is a clear interrelation, which *Conservation Principles* itself underlines, and which flows from that document's adoption of the *Burra Charter's* use of 'place' as a richer and more-than-tangible term than 'building', 'site', 'monument', etc. ([6], e.g., pp. 14–15, 38). Similarly, the UK's overarching *National Planning Policy Framework* (*NPPF*) speaks of 'the desirability of sustaining and *enhancing* the significance of heritage assets' ([8], emphasis added).

This question of how change to historic buildings should be regarded has been a constant concern for conservation from its inception. Under his ICOMOS presidency from 2008 to 2016, Gustavo Araoz spurred an international debate over a 'New Heritage Paradigm', central to which is what he termed the historic environment's 'tolerance for change'. In a paper delivered to the ICOMOS Advisory Committee in Valletta in October 2009, he characterised the development of conservation thus:

> During the 19th and most of the 20th century, the heritage conservation community developed under the assumption that all values attributed to places rested on the material evidence of the place. Thus, the theory and praxis of conservation evolved [...] as an increasingly sophisticated effort *to prevent form and space from undergoing changes* ([9], emphasis added).

This argument was played out in two ICOMOS conferences in 2010 and 2011, including a version of Araoz's paper and a robust response from his predecessor as President, Michael Petzet [10,11]. Petzet was particularly keen to challenge the idea of conservation as the management of change, describing this as an 'inconsiderate [unconsidered?] general proposal', and warning that:

> the core ideology of our organization is being counteracted. After all, conservation does not mean 'managing change' but preserving—preserving, not altering and destroying: ICOMOS [. . .] is certainly not an International Council on Managing Change [12,13].

Petzet also took critical aim at the *Burra Charter*, controversially dismissing this as 'an "Australian" heritage philosophy [which] is quite confusing and suitable for damaging the traditional objectives of monument conservation' ([13], pp. 10–11).

That debate overlapped with the development of the *Recommendation on the Historic Urban Landscape* (HUL) [14], and Araoz specifically cites historic urban areas in support of his argument: '. . .an important cultural value of the historic city rests precisely upon its ability to be in a constant evolution . . .' ([9], p. 58). HUL, interestingly, is an aspect of international conservation almost entirely ignored by Historic England [15] [2].

From this selective overview, it is clear that the approach to change in conservation can be said to be significantly contested, as much internationally as in the UK. On the one hand, Historic England's overarching guidance borrows and develops the *Burra Charter's* fourfold values structure, according the non-tangible (in the form of communal value) a central role in the heritage process. On the other hand, the implications of this broadening of heritage are by no means universally accepted, with much of the discourse around heritage continuing to be argued from a tangible-centric and preservationist position.

The argument between Petzet and Araoz is at first sight perplexing. How can two such prominent heritage professionals—successive presidents of the leading international conservation body, no less—espouse such different approaches? To begin to account for this, it is essential to acknowledge that conservation does not operate only at the level of the explicit knowledge contained in closely worded conservation charters, protocols, and guidance—which together form the discipline's 'Doctrinal Texts'. Like any human endeavour, conservation is also replete with *implicit* knowledge—the underlying structure that scientist and philosopher Michael Polanyi termed the 'tacit dimension' [16]. This feeds and shapes our understanding, and thus the knowledge we articulate, and the processes by which we organise our discipline. Noting the role played by this 'tacit dimension' is by no means a criticism; rather, it shows that conservation is a living tradition, that is, an ongoing, intergenerational, communal argument over what constitutes the good [17]. However, this brings into play such fundamental questions as our understanding of the purpose of the discipline—an explicit concern for Petzet—and what good practice looks like. It is at that foundational level that the course of conservation's original development remains pertinent and active.

Both individual buildings, and conservation as a whole, are contested. In 1906, the novelist Thomas Hardy (who, before becoming a writer, had himself been an architect) presented a paper to the SPAB annual meeting, entitled 'Memories of Church Restoration' [18]. Hardy stresses the contested character of historic churches, and merits quoting at length:

> At first sight it seems an easy matter to preserve an old building without hurting its character. Let nobody form an opinion on that point who has never had an old building to preserve.
>
> In respect of church conservation, the difficulty we encounter on the threshold, and one which besets us at every turn, is the fact that the building is beheld in two contradictory lights, and required for two incompatible purposes. To the incumbent the church is a workshop or laboratory; to the antiquary it is a relic. To the parish it is a utility; to the outsider a luxury. How [to] unite these incompatibles? A utilitarian machine has naturally to be kept going, so that it may continue to discharge its original functions; an antiquarian specimen has to be preserved without making good even its worst deficiencies. The quaintly carved seat that a touch will damage has to be sat in, the frameless doors with the queer old locks and hinges have to keep out draughts, the bells whose shaking endangers the graceful steeple have to be rung.
>
> If the ruinous church could be enclosed in a crystal palace, covering it to the weathercock from rain and wind, and a new church be built alongside for services (assuming the parish to retain sufficient earnest-mindedness to desire them), the method would be an ideal one. But even a parish entirely composed of opulent

> members of this Society would be staggered by such an undertaking. No: all that can be done is of the nature of compromise ([18], pp. 204–205).

Hardy is clear that this is a difficult issue. He characterises historic churches as contested between incumbent, antiquary, parish, and outsider. This plurality boils down to viewing the building 'in two contradictory lights', as something 'required for two incompatible purposes'. He labels these two purposes 'utilitarian machine' and 'antiquarian specimen', respectively, presaging the contemporary distinction between intangible and tangible, and firmly placing them in mutual opposition. His 'crystal palace' ideal is preservationism in its purest (and purist) form—the museumification of tangible heritage by literally placing the historic building in a glass box. Clearly, such an approach would be unfeasibly expensive, something Hardy immediately acknowledges. However, even were it feasible, it would be far from 'ideal'; indeed, I suggest it would be disastrous for the heritage such a building constitutes, holistically understood.

It is worth noting that this statement of the preservationist ideal may not only suit the latter-day antiquarian resolutely focused on the tangible, but in today's context potentially also suits those who see heritage as essentially intangible. Why? Because both ignore the interconnection of people and place. Both also tend to ignore the recognised importance of retaining buildings in beneficial use ([6], p. 43). Without an adequate account of how tangible and non-tangible are integrated, it is easy to argue that a community would be better able to express their intangible heritage if 'freed' of the 'burden' of an awkward historic building. This is precisely what Hardy, in older language, offers as his ideal, and also what some current church communities indeed long for; both, in my view, profoundly misread the nature and importance of tangible heritage.

Hardy's essay remains highly relevant, not only for his diagnosis of the contested nature of heritage, nor only for underlining the link between antiquarianism and preservation, nor again only for foregrounding the tension between the tangible/'antiquarian' and intangible/'utilitarian'. In addition to all of these, the relevance of Hardy's essay lies in demonstrating the continued influence of the preservationist approach as an animating force underpinning the mindset of many conservation practitioners.

From its antiquarian foundation, conservation has developed an art historical approach to historic buildings, which arguably still predominates. Clearly, it would not be possible for conservation practitioners to function responsibly without attending to the art historical aspects of a given building—without the art historical, one would be unable to situate the building in its context, to read its development, or to judge the building's capacity for change. But, however *necessary*, we should not fool ourselves that art history is, of itself, *sufficient*. We must recognise that all art history is written from a particular view, with a particular agenda—as an example, Zachary Stewart provides an excellent demonstration of this in a recent paper on the differing readings of the development of the Perpendicular style from the late fourteenth century onwards [19].

Not only can an art historical approach never be neutral, it inevitably brings with it a tendency to read historic buildings as completed artworks. The philosopher Hans-Georg Gadamer addresses this very issue, insisting that:

> A building is never only a work of art. Its purpose, through which it belongs in the context of life, cannot be separated from it without its losing some of its reality. If it has become merely an object of aesthetic consciousness, then it has merely a shadowy reality and lives a distorted life only in the degenerate form of a tourist attraction or a subject for photography. The "work of art in itself" proves to be a pure abstraction [20].

The abstraction of the 'work of art in itself' flies in the face of the reality of most historic buildings, which typically have changed multiple times over their history to date, and which, through their continuing purpose, stubbornly continue to 'belong in the context of life'. Some 30 years ago, Stewart Brand persuasively made the functional case that it is in the nature of buildings to change with his classic book *How Buildings Learn: What Happens After*

*They're Built* [21]. Furthermore, that history of change becomes an integral and inseparable part of their character, such that when we encounter historic buildings, it is often precisely for that unfolding, messy, and hybrid nature that we love them. Many historic buildings owe their very survival to that ability to adapt to changing needs; without changing, many of these buildings would have been discarded. Furthermore, their continuing use—whether for their core purpose or following adaptive reuse—will lead to further pressure for change. This is not to be regretted, but instead is a sign of health, of a strong link between people and place. Indeed, Historic England recognises this reality, stating that 'Keeping a significant place in use is likely to require continual adaptation and change' ([6], p.43).

## 3. Subtractive Significance and Critical Approaches

By contrast, approaching historic buildings as completed artworks effectively forecloses change. In the art world, conservationists rightly agonise over whether and how to restore an old artwork, and there are examples of parallel discussions in building conservation [22,23]. Few would credibly think they could propose elective change to a Rembrandt without compromising or destroying its integrity, its authenticity, its value (though this is precisely the issue Chinese artist Ai Weiwei has explored with his use of 2000-year-old Chinese vases, in self-descriptive works such as *Dropping a Han Dynasty Urn* (1995), and *Han Jar Overpainted with Coca-Cola Logo* (1995) [24,25]). We can describe this building-as-artwork model as displaying a 'subtractive' understanding of significance—that the work starts from a more or less maximal state of significance, and change can only reduce that bank of significance.

This view of significance as subtractive can be traced from the earliest days of conservation and remains dominant among many conservation professionals to this day. It is evident in the words used in official heritage writing; for example, the preambles to the World Heritage Convention has much to say about threat, harm, and loss:

> Noting that the cultural heritage and the natural heritage are increasingly *threatened* with *destruction* not only by the traditional causes of *decay*, but also by changing social and economic conditions which aggravate the situation with even more formidable phenomena of *damage* or *destruction*,

> Considering that *deterioration* or *disappearance* of any item of the cultural or natural heritage constitutes a *harmful impoverishment* of the heritage of all the nations of the world [. . .]

> Considering [. . .] the importance, for all the peoples of the world, of safeguarding this *unique* and *irreplaceable* property. . . ([26], emphasis added).

Familiar as it may be, the subtractive understanding tends towards absurdity when applied to the less-than-complete loss that results from change to historic buildings in use. If change can only harm significance, then a building such as a medieval church which will often have undergone a dozen or more episodes of change—at times including the removal of whole sections alongside alterations and additions—must have precious little significance left. Clearly that is not the case, because we more often value these buildings precisely *because of* those episodes of change; rather than lamenting each particular loss, we see benefit in the whole.

There must, therefore, be more going on than mere subtraction. Where previous episodes of change to these buildings were 'successful', it was because they took place within a tradition. William Morris recognised this in his celebrated 1877 manifesto for the Society for the Protection of Ancient Buildings, to which modern conservation in the UK generally dates its inception:

> a church of the eleventh century might be added to or altered in the twelfth, thirteenth, fourteenth, fifteenth, sixteenth, or even the seventeenth or eighteenth centuries; but every change, whatever history it destroyed, left history in the gap, and was alive with the spirit of the deeds done midst its fashioning [27].

Morris's claim is that such change took place in a bygone and irretrievable age. He regards historic buildings 'as monuments of a bygone art, created by bygone manners', pleading

with us 'to remember how much is gone of the religion, thought and manners of time past' [27].

In this way, Morris, the great medievalist and champion of ancient buildings, declares himself a thoroughgoing modern. In a more critical mode, anthropologist Bruno Latour explicitly links this understanding of modernity—as following a definitive rupture with the past—with conservation:

> As Nietzsche observed long ago, the moderns suffer from the illness of historicism. They want to keep everything, date everything, because they think they have definitively broken with their past. The more they accumulate revolutions, the more they save; the more they capitalize, the more they put on display in museums. Maniacal destruction is counterbalanced by an equally maniacal conservation [28].

The commitment to modernity and its claimed break with the past—clearly evident in both Morris and Hardy after him—lies at the heart of the preservationist approach. How ironic, then, that we think it legitimate to entrust historic buildings (many dating from before the dawn of modernity) to a system that not only was developed in response to the excesses of modernity but—if Latour is right—is itself a direct expression of those excesses.

In several papers, Cornelius Holtorf has critiqued the 'conservation paradigm' which sees heritage as a finite and diminishing resource that must be defended at all costs, suggesting, provocatively, that 'destruction and loss are not the opposite of heritage but constitutive of it' [29–31]. In his 2015 paper, he applies the insights of loss aversion theory (from the field of economics) to cultural heritage, critiquing the longstanding preference in Western cultural heritage for avoiding loss over acquiring gains of the same value. He uses examples of wholesale loss—including the Fantoft Stave Church, Norway (destroyed by arson in 1992), the Gloucester home of murderers Fred and Rose West (demolished by the authorities), and Ai Weiwei's destruction of ancient Chinese vases noted above—to show that the significance of heritage can persist and indeed grow after their destruction. Holtorf's examples, and others like them, flatly contradict the subtractive understanding of significance.

Similarly, Rodney Harrison describes how heritage practice in the West at least up until the late twentieth century focused on the collection of remarkable buildings, terming this 'a *canonical* model of heritage' ([32], emphasis original). On this view, heritage is separated from and deemed more valuable than the present and its cultural production, and contrasts with more recent models he characterises as 'continuous'. As Harrison discusses, the adoption by conservation of the idea of the cultural landscape in response to the concerns of Indigenous peoples speaks powerfully of the interrelation of tangible and non-tangible ([32], pp. 114–139). As Graham Fairclough points out, the idea also has profound implications for the understanding of continuity and change:

> The idea of cultural landscape has the concept of change (in the future as well as in the past) at its very heart. The idea that there are any landscapes where time has stood still, and history has ended, is very strange. No landscape, whether urban or rural, has stopped its evolution, no landscape is relict: it is all continuing and ongoing [. . .]. The decision that each generation, including archaeologists has to make, is what will happen next to the landscape, and how it will be managed or changed [33].

The development in the last two decades of Historic Urban Landscape thinking, from the 2005 Vienna Memorandum to the 2011 Recommendations and beyond [14,34,35], has brought with it a holistic approach and a blurring of the boundaries between building, context and urban area [36]. As suggested by Gustavo Araoz's engagement with HUL (noted above), we can in time expect insights from the field of cultural landscape to inform conservation, to its enrichment.

That said, with the notable exception of Araoz, none of these mentioned thinkers who are challenging the preservationist 'conservation paradigm' comes from amongst

practising conservation professionals. Indeed, most are archaeologists by background. The architects and surveyors who become conservation specialists often do so without further academic qualification, and even if they do postgraduate qualifications, cannot be guaranteed to encounter heritage studies thinking of the sort represented by Holtorf, Harrison, and Fairclough [37] [3].

## 4. Church Buildings

We turn next to an oddity of British heritage protection, rooted partly in pre-modernity, that is suggestive of a different understanding of the relation of tangible and non-tangible heritage.

The Ecclesiastical Exemption offers a parallel consent mechanism within UK law for Christian places of worship that remain in use; while not unique, this is unusual in international terms [4]. The Exemption is an administrative umbrella, a set of criteria within which individual Christian denominations can develop their own systems to control alterations to listed buildings. It is an exemption from listed building consent only, not from planning permission, nor from Scheduled Monument Consent, nor indeed from Building Regulations approval. The Exemption applies in different forms in each of the four nations of the UK; in England, which has the greatest number of protected buildings, the Exemption covers five denominations—the Church of England, the Roman Catholic Church, the Methodist Church, the United Reformed Church, and churches within the Baptist Union.

In 2010, the UK's Department of Culture, Media, and Sport (DCMS) published guidance to accompany the last revision to the legislation [38]. This guidance insists that any procedures under the Exemption 'must be as stringent as the procedures required under the secular heritage protection system'; this 'equivalence of protection' is identified as a key principle, and one which will be kept under review to ensure that appropriate standards of protection are maintained ([38], p. 7). There is always, therefore, the potential for the Exemption to be withdrawn, or indeed for a denomination itself to withdraw from it, as did the United Reformed Church in Wales in 2018.

The guidance appreciates the importance of keeping historic buildings in use, if they are to survive. It states:

> The Ecclesiastical Exemption reduces burdens on the planning system while maintaining an appropriate level of protection and reflecting the particular need of listed buildings in use as places of worship to be able to adapt to changing needs over time to ensure their survival in their intended use ([38], p.6).

This offers explicit recognition that, for living listed buildings such as churches, change is legitimate in principle, and essential to their survival as places of worship; at the same time, it points towards the brutal reality that, should the Exemption be withdrawn, the secular planning system would simply be unable to cope.

The Church of England has by far the greatest number of listed churches (some 12,300), of which 4300 are listed grade I and a further 4300 grade II*. Because of both the numbers and the highly listed nature of the buildings in question, the Church of England's Faculty Jurisdiction is the most developed system under the Exemption. The system dates from 1913, in the early days of heritage protection in England, and was built on the Church's existing faculty system, which dates back to medieval times, arguably, therefore, helpfully bringing with it elements of pre-modernity. As a result of this genesis, the system is legal in nature, with each diocese having a 'consistory court', presided over by a judge, known as the 'chancellor'. Contested decisions are set down in publicly available judgments [5]; these record the arguments in the case, and combine to form an invaluable research resource, as well as helping satisfy the Exemption requirement for openness and transparency.

Each diocese is required to maintain a Diocesan Advisory Committee (DAC), which should cover specific areas of expertise, including in the development and use of church buildings, in liturgy and worship, in architecture and archaeology, and in the care of historic buildings and their contents. DACs (and equivalent bodies in other exempt denominations)

thus bring together a far wider range of specialist skills than the secular system has at its disposal, offering a vital source of pro bono expert advice.

While it goes to great lengths to mirror the rigour of the secular process, and to satisfy the requirement for openness and transparency, it is freely acknowledged that the Faculty Jurisdiction will often produce different results from the secular system. Because of this, some UK conservation professionals regard the Exemption as an aberration, giving unwarranted and unwelcome licence to the major church denominations to make their own rules, leading to the degrading and potential destruction of some of the nation's most important heritage. However, for the Exemption's defenders, the secular system is both ill-equipped and incapable of dealing adequately with the particularity of churches; on that latter view, a separate system is essential because the secular system does not understand the non-tangible aspects of heritage inherent in a church building. How, for example, could a secular conservation officer begin to assess the justification for liturgical changes such as a reordering of furniture, the introduction of a nave altar or the moving of a baptismal font, without themselves being conversant with such buildings in use?

One landmark decision is the 2012 appeal judgment over St Alkmund, Duffield, which now plays a central role in the Faculty system, providing an established framework for guiding chancellors in their judgments, and acting as a common point of reference for all parties ([39]; for a fuller discussion of this judgment, see [40]). It also powerfully illustrates how non-expert voices can be closely attended to alongside the expert, providing evidence of a welcome rebalancing of that sometimes-fraught relationship. The Duffield judgment, along with many others like it, suggests that the Exemption is an example of an official conservation process that recognises the reality of living heritage. While it may be unusual, the Exemption is thus highly significant—perhaps as some form or prototype or forerunner—in the present context of the shift in heritage theory. Further, the Exemption demonstrates that this shift need not be *away from* the tangible, but instead *towards* a holistic understanding of heritage as a nexus—literally a binding together—of tangible and non-tangible.

A second demonstration of a growing appreciation of non-tangible value and community participation in heritage is offered by some recent research by the National Churches Trust (NCT), a grant-making and campaigning charity; that research aimed to quantify the economic and social value of church buildings to the UK [41] [6]. The resulting *House of Good* report is based on HM Treasury's official Green Book methodology—the government's own means of assessing social value—and concludes that the annual contribution of church buildings to the UK economy is an extraordinary 55 billion GBP. Given that there are estimated to be just under 40,000 places of Christian worship in use in the UK, that is an average of 1.4 million GBP for each and every church building. Over three quarters of that astounding total lies in the wellbeing value to individuals from community activities run or hosted by churches, such as food banks, youth groups, drug and alcohol support, etc.

Naturally, not all of those 40,000 churches are buildings of tangible heritage significance, though perhaps 40% are, including, as we have seen, just under half of all of England's grade I listed buildings. Furthermore, the sorts of activities that attract such high levels of social value in the report take place in buildings across the full range of heritage value.

Clearly, this sort of financial methodology—designed to help central government compare policy priorities—can never deliver a complete description of the rich significance of historic church buildings. However, while no one would claim the description to be all-encompassing, it surely does form a necessary part of any balanced assessment. Heritage voices from a conservative position that might argue that food banks, etc., are at best incidental to the life of a historic church building miss something vital (literally) about living buildings: that such activities represent a continuation of a tradition that has been a part of this heritage—the amalgam of people and place—from their inception. Furthermore, these non-religious, social activities are also highly relevant to the question at hand; such

research demonstrates plainly that there are categories other than the purely material that must be considered when determining the importance of historic buildings in use.

## 5. The Centrality of Living Buildings

The startling figures in the NCT research raise the issue of 'living buildings', what is meant by that evocative term, and what implications this may have for conservation as a whole. The distinction of 'living' from 'dead' buildings dates from the early days of heritage protection, for example featuring in the first three of six resolutions of the Sixth International Congress of Architects held in Madrid in 1904. The first reads:

> Monuments may be divided into two classes, *dead monuments*, i.e. those belonging to a past civilization or serving obsolete purposes, and *living monuments*, i.e. those which continue to serve the purposes for which they were originally intended ([42], emphasis original).

The second resolution states that 'Dead monuments should be preserved . . .', while the third says that 'Living monuments ought to be restored so that they may continue to be of use, for in architecture utility is one of the bases of beauty.' It is the fact that these buildings continue in use that is seen as so pivotal, and the last comment on the relation of beauty and utility is commensurate with Gadamer's view of buildings discussed in Section 2 above.

Church buildings can be termed 'living' in at least three, typically overlapping, senses: first, in almost all cases, these buildings continue in the use for which they were first built; second, their principal users are a community, previous generations of which generally created the building in the first place and whose continuous presence has been at the heart of the building ever since; and, third, the building has typically changed multiple times through its history, thus being more dynamic than static in nature.

Veteran conservation architect Donald Insall chose *Living Buildings* for the title for a retrospective monograph covering 50 years of practice; he argues that all historic buildings are 'alive and constantly changing', whether through successive intervention or the effects of time and weather [43]. In an early chapter entitled 'Buildings are Alive', he offers the example of St Anne's Church, Kew which, since 1714, has evolved through nine distinct stages ([43], pp. 42–43). If, as Insall suggests, all buildings are alive in this way—at least to some degree—then this poses fundamental questions for conservation, and rules out any notion that the purpose of conservation might be 'to keep things the same'.

And yet, I suggest that for as long as the discipline clings to its preservationist roots—as exemplified in Holtorf's 'conservation paradigm' and Hardy's comment in Section 2—preservation will remain its basic orientation, feeding a residual bias against change, however judicious. This shows in the use of language, e.g., in the frequent use of the word 'harm' where 'change' or 'impact' would be more appropriate, as touched on in Section 3 above. Clearly, *some* change can be extremely harmful to heritage, as the discipline knows to its cost, but that is not true of *all* change. The view that all change to historic buildings is harmful is one that can be argued, but it is far from neutral, being heavily invested in that preservationist understanding of heritage discussed earlier.

The understanding that buildings grow and acquire identity as they do so is paralleled by social anthropologist Tim Ingold, known for his phenomenological approach to the relation of people to their material environment. Ingold questions the dominant 'genealogical model' within archaeology, 'namely that persons and things are virtually constituted, independently and in advance of their material instantiation in the lifeworld' [44]. This he roots in Aristotelian hylomorphism, the belief that physical entities are compounds of matter and immaterial form (for more of Ingold's critique of hylomorphism, see [45]). In place of this understanding of people and things as 'created in advance', he suggests it is instead better 'to think of a world not of finished entities, each of which can be attributed to a novel conception, but of processes that are *continually carrying on*, and of forms as the more or less durable envelopes or crystallisations of these processes' ([44], p. 163, emphasis original). He helpfully critiques traditional archaeology as 'too concerned with that which has been preserved in the archaeological record, which he characterises as fragments that have

> broken off from the flow of time, [and that] recede ever further from the horizon of the present. They become older and older, held fast to the moment, while the rest of the world moves on. But by the same token, the things of the archaeological record do not persist. For whatever persists carries on, advancing on the cusp of time ([44], p. 164).

In language very similar to that used by Gadamer above, this notion that only those things that advance 'on the cusp of time' persist is a radical restatement of the accepted conservation wisdom that continued beneficial use of historical buildings is desirable ([6], p. 43; see also [46]). The parallels with living buildings are clear; Ingold's restatement is highly relevant to living buildings.

If living buildings are moved from the status of being a special case to being seen as the norm—as Ingold's argument would imply, and Insall suggests they should—then this would have profound implications for conservation practice, particularly in the need to do the urgent work of developing a more robust theoretical foundation for the discipline. Important work has already been done to that end, including under the Living Heritage Sites Programme at the 'International Centre for the Study of the Preservation and Restoration of Cultural Property' (ICCROM). Ioannis Poulios, in his important study of the monastic sites at Meteora in Greece, has identified four forms of continuity that sustain living heritage: continuity of function, of relation between a core community and the tangible heritage, of expressions (entailing a recognition that heritage places will continue to change), and of care [47]. To be clear, continuity here refers to dynamic continuity of an ongoing process, not the static continuity of an unchanging product. And that third form of continuity, of expression, entails ongoing creativity, something Poulios foregrounded in an earlier paper [48]. A similar fourfold continuity is present in the excellent paper *Living Heritage: A Summary*, published by ICCROM [49].

We can note, therefore, that a living heritage approach is distinct from much conventional Western conservation thinking, in two respects of particular relevance to this discussion. First, a living heritage approach acknowledges the principle of the legitimacy of change. Second, just as the community across time needs to be considered as a whole—by placing it within the ongoing and intergenerational argument that is its tradition—so too the building itself should be valued for the open-ended holism that encompasses all its stages of development, not only for the current stage at which the story happens to have paused for a time. That is a temporal perspective that comes from operating within an active tradition. In the case of church buildings, this would be the Christian tradition, central to which is the idea of the Communion of Saints—that is, that the Church is a community extended through time, including those generations long past, those currently alive and, by extension, those still to come [50]. When dealing with buildings of that tradition, it is surely more appropriate to deal with them in this way, rather than as modernity does, as specimens of a bygone art, as we saw Morris suggesting above.

Conservation is faced with a choice—and one that will define it for generations to come—over whether or not to engage with living heritage, and allow itself to be changed by it. It will be very attractive to hunker down in the tangible realm, but this requires discounting the relevance of those community voices who, in attempting to live well with their buildings, seek to change them. Or, conservation can join Heritage Studies in attending to those community voices, embracing living buildings and welcoming the enrichment of the discipline that will result. The outcome of that positive turn would be more change to historic buildings, but also more historic buildings in better condition and with better prospects of surviving long into the future. And that, it should be stressed, should *not* be seen as a trade-off, tolerating change and even loss to achieve the benefit of survival, as with the Fantoft stave church and other extreme examples discussed above. Rather, a living buildings perspective puts *both* factors on the positive side of the balance.

For conservation to respond positively in the way described will require it to undertake its own theoretical work. In particular, this will involve a recognition of living buildings as central rather than peripheral both to conservation and to heritage more broadly. It will

also require reflection on the nature of change, and from that the development of guidelines and processes for distinguishing change that causes unacceptable harm, from change that enriches a building by continuing its narrative. Conservation's current processes are not adequate to that task. They must be rethought, which will involve tracing through the implications of choosing a different starting point from that of the preservationists who so effectively set the agenda for the early stages of conservation's development.

## 6. 'Generative' Significance

What, then, are the implications for our understanding of significance? At the start of this paper, the current framing of significance was characterised as 'subtractive', based on a predominantly art historical understanding of built heritage. The Ecclesiastical Exemption, the *House of Good* report, the living building approach, and other heritage voices each in their different ways illustrate the limitations of such an understanding. This final section considers what an alternative understanding of significance might look like.

Given that this theoretical work is done in the context of recognising the creative and ongoing nature of living heritage, we could provisionally label this alternative view of significance as 'generative'. In practice, many conservation professionals rarely engage with theory, instead seeing the aesthetic, the historical, and the techniques of material conservation as the core of the discipline; this priority is reflected in the 14 criteria listed in the ICOMOS *Guidelines for Education and Training* [51]. And yet, this theoretical work is unavoidable. Even if we wanted to avoid theory, we cannot—one either engages with it deliberately, or one finds oneself manipulated by the theory of others.

It should be restated that this paper is not arguing against the discipline of architectural history or, more generally, art history. The art historical will be a prominent concern in most cases of physical heritage. We can say that it will always be a necessary part of conservation, but of itself it can never be sufficient. And that goes back to our definition of the heritage we are attempting to conserve—not place (tangible) alone, nor indeed people (intangible) alone, but a hybrid or an amalgam or a binding together of the two. The issue is that unless the foundational understanding that sees conservation in predominantly art historical terms is directly challenged, the subtractive understanding of significance will endure. From that source flows a conservation system built around resistance to change, at the attendant cost of much heritage being 'lost in translation'.

Under a 'generative' view of significance, by contrast, the identity of a historic building is not seen as something fixed, but as something that continues to develop. It remains possible that the building may be harmed by proposed change but, equally, with the right approach and craft skills, its significance can be enhanced. Such a view of significance would be a much better fit for the sort of living buildings described above, buildings with a long history of change. And in an English context, such a view fits well with government policy, which requires local planning authorities to take account of 'the *desirability* of sustaining and enhancing the significance of heritage assets and putting them to viable uses consistent with their conservation' [8].

The generative view also fits well with the *Burra Charter* (and, by extension, with parts at least of heritage policy in the UK). Steve Brown notes that the changing face of heritage management can be traced through the versions of the *Burra Charter*, away from the idea of the importance of a heritage asset as something 'inherent, immutable or somehow "fixed" within it' [52]. For the *Burra Charter* (and those that draw on it), this importance is termed 'cultural significance'; if, as envisaged, local communities are to have any role in the assessment of that significance, then both the assets themselves and their significance can be expected to change—to be 'mutable' rather than 'mute', to use Brown's terms.

Dirk Spennemann has taken the idea of 'shifting baseline syndrome' from the field of historic ecology and has applied it to heritage studies to explore how the passage of time and intergenerational change impacts the assessment of heritage significance, concluding that significance will wax and wane [53]. It follows that, over time, the mutability of significance may come into tension with the original reasons for inscription on a protected

list, exposing unavoidable biases; this is readily visible in the English listing system, the initial phases of which were done at pace from 1944 onwards and which largely covered medieval churches, country houses, and buildings from before 1750 [54].

It seems no accident that many of the authors concerned with change and heritage draw on Australian experience, including of course Laurajane Smith [55], as well as Brown, Spennemann, Harrison, and many others. As Brown points out, the Australian context demands heritage processes that encompass Indigenous as well as more conventional 'European' heritage, generally in the form of historic buildings and sites ([52], p. 21). This reflects the Araoz–Petzet divide discussed in Section 2 above; the inclusion of the UK on the change side of the argument is due to Historic England's adaptation of the Burra framework in *Conservation Principles*, which is explicitly referenced by Petzet in his inclusion of the UK alongside Australia and the USA ([12], p. 54; [13], p. 10).

Furthermore, the generative view of significance also fits well with the narrative approach mentioned above, which stresses the ongoing development of historic buildings [11,56]. On this view, a building's significance is always provisional, because we do not know where its story will go in future—to claim otherwise is to put too great an emphasis on the past and present, on where the story happens to have reached now.

In her 2017 Reith lectures, the late historical novelist Hilary Mantel provocatively suggested that:

> History is not the past—it is the method we have evolved of organising our ignorance of the past. It's the plan of the positions taken, when we stop the dance to note them down. It's what's left in the sieve when the centuries have run through it. It is no more "the past" than a birth certificate is a birth, or a script is a performance, or a map is a journey [57].

Mantel's argument, of course, is against a reification of the past into something fixed and fully defined, echoing Ingold's critique, as discussed above; her fourfold opposition of fixed form (plan/birth certificate/script/map) to lived reality (dance/birth/performance/journey) is as compelling as it is lyrical. I suggest that the same can be claimed, perhaps more strongly still, of attempts to codify the significance of historic buildings that continue to change. A 'generative' view of significance still sees the current state of a historic building—in Mantel's terms 'what's left in the sieve when the centuries have run through it'—as hugely important. It treats the building's history with the utmost respect—after all, how can one understand a story without understanding how it began and developed to this point? However, where the 'subtractive' view treats this as all that matters, the 'generative' view treats any assessment of 'significance to date' as provisional, partial, and incomplete.

It is certainly not being argued that change is a good thing in and of itself—that might have been a core belief of high modernity, but the emptiness of its blind faith in progress is all too evident, and our historic environment bears the scars of such naivety. Rather, the argument is that change should not be ruled out *a priori*—as is the preservationist position, and one seemingly shared by Michael Petzet, as noted above. The question of how historic buildings should be allowed to change is far more complex than a simple question of yes or no. Rather, the task is to decide how a building can be changed without destroying its narrative coherence and integrity, for which a more holistic understanding of significance is required than the 'subtractive'.

Once the legitimacy of change to historic buildings is accepted in principle, the key question becomes what sorts of change should be allowed. Given that each historic building is different, that question will resist any top-down answer. Instead, I have elsewhere proposed that historic buildings (of any age) can more helpfully be considered in terms of an ongoing narrative rather than as a completed artwork [11]. In the case of a church, that is a community narrative written over multiple generations of the core community, and one that awaits further chapters. Once narrative is accepted as the central metaphor, the question of managing change is transformed from attempting to keep change to a minimum to considering how to extend the story well, how to add a worthy new chapter to the existing narrative, when one is called for. In order to do that responsibly, we need

to understand the story to date as well as we possibly can, but also to be bold with our current chapter, while having the humility to acknowledge that future generations will wish to add their own chapters.

## 7. Conclusions

This paper has sought to use the example of historic English Parish Churches and their care and conservation under the Ecclesiastical Exemption to illustrate the shift away from heritage understood as tangible product towards an understanding of heritage as non-tangible process. The Ecclesiastical Exemption has been presented as one expression of a living buildings approach, with the suggestion that a 'subtractive' understanding of significance should be replaced by a 'generative' model. On this latter view, heritage as a whole is understood as an amalgam/hybrid/nexus of tangible and non-tangible—a whole that resists analysis into its parts. The implication for conservation specifically is that the 'living' status of historic buildings is seen as central, not peripheral; to respond positively to this reality requires conservation to reappraise its own narrative, both its genesis and its future direction.

The alternative—defending the preservationist 'conservation paradigm' and avoiding such a reappraisal—threatens significant harm to heritage. By deliberately making change difficult, one of two things can happen: first, living buildings become ossified, ultimately destroying the heritage lying in the joining of tangible and non-tangible; or, second, there is a flight from the material through an understanding of intangible heritage that denies our inherent physicality and the enduring importance of tangible heritage. Conservation is in urgent need of thorough engagement with the non-tangible aspects of heritage; the first step will be to engage with the living nature of tangible heritage.

**Funding:** This research received no external funding.

**Institutional Review Board Statement:** Not applicable.

**Informed Consent Statement:** Not applicable.

**Data Availability Statement:** Not applicable.

**Conflicts of Interest:** The author is currently a trustee of the National Churches Trust.

## Notes

1. The proposed document was issued for consultation, but never formally published; it would have moved policy away from international principles such as the *Burra Charter*, including demoting communal value to become a subset of historic value.
2. On its reception in England, and the relevance of HUL for conservation as a whole, see [15].
3. The University of York's MA in Conservation of Historic Buildings, now running for 50 years, offers one example of postgraduate study in an archaeology context; for a vision of Conservation without Heritage Studies, see John Earl's misnamed but otherwise excellent [37].
4. It is interesting to note that Malta, for example, combines aspects of British and Italian practice, and has its own version of the Exemption.
5. There is a Searchable Database of Judgments at 'Judgments Index'. Available online: https://www.ecclesiasticallawassociation.org.uk/index.php/judgements/judgments-a-z (accessed on 21 June 2023).
6. The 2021 Update is the source for the figures that follow.

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
