# Peer review of "Lost in Translation: Tangible and Non-Tangible in Conservation"

_2673-8945, doi:10.3390/architecture3030031_

Round 1
Reviewer 1 Report
Thanks for the paper. It is an interesting contribution to an important topic. However, in my opinion, some changes are needed so that it can be better grounded. Here is the list of my suggestions:
_The paper could benefit from a more detailed explanation of what the author means by tangible/non-tangible. Both concepts are inherently ambiguous; I believe clarification is needed. For instance, on page 3 (lines 141-142), the author refers to “the tension between the tangible/‘antiquarian’ and intangible/‘utilitarian’. ” However, it could be argued that function and utility rely also on some intangible qualities of architectural spaces (their character; the way they fit a specific culture and lifestyle, etc.), while the antiquarian also relies on material and tangible qualities. It might also be beneficial to refer to some existing literature to explain both concepts.
_The paper needs more references. Adding more references could help better situate the paper in contemporary discussion. As the author comments, there is an ongoing debate on conservation or change in monumental buildings (which is indeed decades old). However, most of the research relevant to such debate does not appear in the paper. I suggest the author include those investigations he considers most significant for the research, in order to better situate and support his assessments. Furthermore, this debate has evolved in recent years. New concepts have appeared, such as the idea of adaptive reuse in architectural theory. Possibly, considering some contemporary views of the subject could enrich the research.
_On page 8 (lines 366-370) a study of Iannis Poulis is mentioned, but without proper reference (neither in the text nor in the list of references). I suggest the author revise the text to ensure that all research is properly referenced.
_Some claims are not properly supported. For instance, on page 8 (line 412) the author comments that “many conservation professionals might prefer to avoid theory”. In my opinion, it is important to refer to the work of some authors who argue against theory, or reformulate the sentence. Something similar occurs on page 3 (lines 131-134)
_On page 2 (lines 68 onward) the author refers to an argument between Michael Petzet and Gustavo Araoz. I consider that a more in-depth explanation of this argument could be beneficial.
_A point of concern to me is that in some passages the paper seems to suggest that non-strictly conservative approaches (that is, approaches that contemplate the possibility of implementing modifications to a monument to adapt to changing times and needs) are a novelty (for instance, page 7, lines 350-35; page 9, 463-464). I am not sure that this is the case. While there is certainly a debate on the subject, much theory and many built works argue for the need to adapt monuments while respecting their socio-cultural and architectural value.
Author Response
Thank you for your helpful comments. I have substantially rewritten parts of the paper, hopefully strengthening the argument, and incorporating more references. I have acknowledged, but not attempted to do address, the related question of adaptive reuse, which in my view demands separate treatment.
Reviewer 2 Report
architecture-2496966-peer-review-v1
Lost in Translation: Tangible and Non-Tangible in Conservation
This is a really well written paper that was a joy to read. I laud the author for his work.
What make me wonder, though, is why the paper was submitted to MDPI Architecture (unranked) and not to MDPI Heritage (a Q1 journal) where it would have been a better fit.
The introduction makes some assertions that should see some references.
Line 51 ff Be mindful not to commingle individual heritage items with heritage conservation areas, which have differing levels of protection. Hence it is not surprising that protection regimes in HCAs are more open to change.
In your section on ‘generative significance’ it may pay to consider that ANY heritage significance is underpinned by community (and professional) values that are projected on heritage assets and that by their very nature these are intra- and intergenerationally mutable qualities, see
Bowdler, S. (1984). Archaeology: Proceedings of the 1981 Springwood Conference on Australian Prehistory. In Site Surveys and Significance Assessment in Australian Archaeology (pp. 1-9). Canberra: Department of Prehistory, Research School of Pacific Studies, Australian National University.
Brown, S. (2008). Mute or mutable? Archaeological significance, research and cultural heritage management in Australia. Australian Archaeology, 67(1), 19-30.
Spennemann, D. H. R. (2022). The Shifting Baseline Syndrome and Generational Amnesia in Heritage Studies. Heritage, 5(3), 2007–2027. doi:10.3390/heritage5030105
Some additional ideas / guidance can be derived from the US historic preservation approach regarding technological heritage that is still in use where the heritage dimension is balanced against the fact that such installations are still functioning laboratories where chaging experimental design requires changes to the fabric
https://www.energy.gov/management/articles/assessment-historic-properties-and-preservation-activities-us-department-energy
https://www.asme.org/wwwasmeorg/media/resourcefiles/aboutasme/who%20we%20are/engineering%20history/landmarks/183-wright-field-5-foot-wind-tunnel.pdf
The one other issue that I think also needs to be addressed is adaptive reuse. There is adaptive use in keeping with the function of the building and there is adaptive reuse for other functions. This can be a death by a thousand cuts and not living heritage.
examples see
Lueg, R. (2011). Houses of God... or not?! Approaches to the Adaptive Reuse of Churches in Germany and the United States. Lueg, R. (2011). Houses of God... or not?! Approaches to the Adaptive Reuse of Churches in Germany and the United States.
Velthuis, K., & Spennemann, D. H. R. (2007). The future of Defunct Religious Buildings: Dutch Approaches to their Adaptive Reuse. Cultural trends, 16(1), 43–66. doi:10.1080/09548960601106979
Amayu, E. (2014). New Uses for Old Churches: An Examination of the Effects of Planning Regulations on the Adaptive Reuse of Church Buildings
and the
Minor Issues
Line 76 typo ‘jn’
Author Response
Thank you for your helpful comments. I have substantially rewritten parts of the paper, hopefully strengthening the argument, and incorporating more references. I have acknowledged, but not attempted to do address, the related question of adaptive reuse, which in my view demands separate treatment. I have also not attempted to address industrial heritage, which again would need some space to do justice to it.
The article was submitted to Architecture by invitation for a special issue.
Round 2
Reviewer 1 Report
Thank you for the paper. In my opinion, it can be published in the present form.
Reviewer 2 Report
The author has significantly revised the manuscript, which is much improved over an already great piece. The author's explanations why certain elements of my review were not addressed are adequate.
I recommend that the paper be published